# Exosomes Derived from Radioresistant Breast Cancer Cells Promote Therapeutic Resistance in Naïve Recipient Cells

**DOI:** 10.3390/jpm11121310

**Published:** 2021-12-06

**Authors:** Chantell Payton, Lisa Y. Pang, Mark Gray, David J. Argyle

**Affiliations:** Roslin Institute, The Royal (Dick) School of Veterinary Studies, The University of Edinburgh, Edinburgh EH25 9RG, UK; chantell.payton@ed.ac.uk (C.P.); mark.gray@ed.ac.uk (M.G.); david.argyle@ed.ac.uk (D.J.A.)

**Keywords:** breast cancer, exosomes, chemoresistance, radioresistance, comparative oncology, One Health

## Abstract

Radiation resistance is a significant challenge in the treatment of breast cancer in humans. Human breast cancer is commonly treated with surgery and adjuvant chemotherapy/radiotherapy, but recurrence and metastasis upon the development of therapy resistance results in treatment failure. Exosomes are extracellular vesicles secreted by most cell types and contain biologically active cargo that, when transferred to recipient cells, can influence the cells’ genome and proteome. We propose that exosomes secreted by radioresistant (RR) cells may be able to disseminate the RR phenotype throughout the tumour. Here, we isolated exosomes from the human breast cancer cell line, MDA-MB-231, and the canine mammary carcinoma cell line, REM134, and their RR counterparts to investigate the effects of exosomes derived from RR cells on non-RR recipient cells. Canine mammary cancer cells lines have previously been shown to be excellent translational models of human breast cancer. This is consistent with our current data showing that exosomes derived from RR cells can increase cell viability and colony formation in naïve recipient cells and increase chemotherapy and radiotherapy resistance, in both species. These results are consistent in cancer stem cell and non-cancer stem cell populations. Significantly, exosomes derived from RR cells increased the tumoursphere-forming ability of recipient cells compared to exosomes derived from non-RR cells. Our results show that exosomes are potential mediators of radiation resistance that could be therapeutically targeted.

## 1. Introduction

Breast cancer is the most common female malignancy and the leading cause of cancer-related deaths in women [1,2]. Similarly, naturally occurring canine mammary tumours are the most common cause of death in intact female dogs and have been proposed as a comparative model of the human disease [3]. Canine mammary tumours have a similar genetic predisposition, histopathology, disease progression and clinical outcome to the human disease. Human breast cancer is commonly classified into molecularly distinct subtypes: normal breast-like, HER2+, luminal A, luminal B and triple negative. These subtypes differ in clinical outcomes, patient survival and treatment strategy. However, there is gene expression heterogeneity within these subtypes and breast cancer can be considered as a spectrum of diseases. Kumar et al., 2012 [4] utilised microarray technology to highlight a 163-gene expression signature associated with prognosis, highlighting that, in the context of gene expression, this disease is highly heterogenous and individualised. Assessing the global gene expression and proteomic profiles of each individual patient and applying that information to a database of available treatment options may be more successful, in terms of survival rates, than following a rigid treatment plan based on tumour subtype [5]. This method of patient-specific therapy assignment would be more efficient in terms of time, expense and patient side effects and may be applicable in both human and veterinary medicine.

The emergence of resistance to key modalities, including chemotherapy and radiotherapy, and the subsequent re-initiation of tumour growth and relapse represent a significant clinical problem, often with limited treatment options and increased mortality. Understanding the underlying molecular mechanisms driving therapy resistance could help to identify potential biomarkers to track the emergence of resistance and novel therapeutic targets.

Tumours comprise a heterogenous mix of cell populations including cancer stem cells (CSCs) and non-CSCs, which make up the bulk of the tumour. CSCs are long-lived cells that drive tumourigenesis as they can self-renew and differentiate into other cellular subtypes. Breast CSCs are inherently resistant to conventional chemotherapy and radiotherapy [6]. Therefore, the relative size of a CSC pool within a tumour may influence the intrinsic radioresistance of that tumour. Radiation treatment will eliminate the majority of cancer cells; however, CSCs will survive and be able to re-initiate tumour growth and tumour cell repopulation leading to patient relapse [7]. The development of acquired therapy resistance can also occur due to selective pressures imposed by cancer therapies that can result in advantageous mutations in newly forming cancer cells and lead to increased survival by, for example, the activation of epithelial-to-mesenchymal transition (EMT), enhanced DNA damage repair and enhanced elimination of cytotoxic content from within the cancer cell [8] including the active chemotherapeutic agents or the reactive oxygen species produced during radiotherapy treatment [9].

Exosomes have been implicated in the acquisition of therapy resistance [10,11]. Exosomes are nanovesicles secreted from most living cells. They have a size range between of 30–150 nm in diameter, and they contain a biologically active cargo consisting of nucleic acids, miRNAs, proteins and lipids, encapsulated within their double membrane [12]. The outer surface of the membrane contains integrins, tetraspanins and cell signalling receptors [13]. The content of exosomes is reflective of the parental cell from which it is derived, and under non-diseased states, the role of exosomes is to mediate cell-to-cell communication [14,15]. As the formation of exosomes within the parental cell results in the incorporation of the contents of the parental cell, the exosome cargo can reflect the development and progression of the diseased state of the parental cell. Further research has shown that the active content of exosomes can result in phenotypic and genotypic changes in recipient cells. In cancer, exosomal transfer can occur between developing cancer cells, and between cancer cells and stromal cells, and can have a range of functions, for example, developing cancer cells can communicate via exosomes to programme stromal cells to provide nourishment in the form of amino acids and carbon [16,17,18]. As well as programming surrounding stromal cells to provide a nurturing environment for cancer cells, exosomes can also promote metastasis and mediate organotropism [19,20].

Exosomes have been shown to play a pivotal role in therapy resistance in humans [21,22], but the role of exosomes in canine therapy resistance has not yet been studied. Exosomes derived from human breast cancer cells have been shown to shuttle chemotherapeutic agents out of the cell [23], and chemotherapy-resistant breast cancer cells can transfer p-glycoprotein protein pumps to chemotherapy-sensitive breast cancer cells to allow the active removal of the chemotherapeutic agents [24]. However, the role of exosomes in the development of radiotherapy resistance in breast cancer cells and the CSC population is poorly understood, and the mechanisms by which exosomes can mediate chemoresistance cannot be directly applied to the development of radioresistance. We hypothesise that exosomes derived from radioresistant (RR) cells can disseminate the RR phenotype to non-RR cancer cells. In this study, we isolated exosomes from the human breast cancer cell line, MDA-MB-231, and the canine mammary carcinoma cell line, REM134, and their RR counterparts to investigate the effects of exosomes derived from RR cells on non-RR recipient cells. Our data show that exosomes derived from RR cells, compared to exosomes derived from non-RR cells, can increase cell viability and colony formation in recipient cells and increase chemotherapy and radiotherapy resistance. These results are consistent in CSC and non-CSC populations. Our results show that exosomes are potential mediators of RR that could be therapeutically targeted. Future research could focus on the profiling the exosomal cargo to identify emerging markers of radioresistance. These biomarkers could be monitored throughout treatment to optimise patient-specific treatment plans for anticancer interventions.

## 2. Materials and Methods

### 2.1. Cell Culture

The cell lines used in this study were the human breast cancer cell line, MDA-MB-231, and the canine mammary carcinoma cell line, REM134 [25]. Radioresistant MDA-MB-231 and REM134 cell lines were gifted by Dr. Mark Gray [26]. RR cell lines were established over several weeks by gradually irradiating the non-RR parental cell lines with increasing doses of Gray (Gy). MDA-MB-231 cells were cultured in Dulbecco’s modified Eagle’s medium (DMEM) + 1 g/L D-glucose, L-glutamine + pyruvate (Gibco Life Technologies, Invitrogen, UK). REM134 cells were grown in DMEM + 4.5 g/L D-glucose, L-glutamine—pyruvate (Gibco Life Technologies, Invitrogen, UK). All cell culture media were supplemented with 10% exosome-depleted FBS and 1% penicillin and streptomycin. Cells were maintained at 37 °C in 5% CO_2_ in a humidified incubator. FBS was depleted of exosomes by ultracentrifugation in an SW32 Ti rotor (Beckman Coulter, IN, USA) at 12,000× *g* for 18 h at 4 °C.

### 2.2. Radiation Treatment

To maintain the RR phenotype, RR cells were irradiated with 12 Gy every 3–4 weeks. Briefly, cells were grown until 70% confluence and, after standard trypsinisation, were resuspended as single cells in 10 mL of the appropriate media and immediately irradiated in the gamma cell irradiator (Gammacell 1000 Elite, Best Theratronics, Ottawa, ON, Canada) in 50 mL falcon tubes. After irradiation, cells were transferred into a T75 flask and maintained as previously described.

### 2.3. Exosome Isolation

Cells were seeded in T175 flasks and grown until 70% confluence. Cells were washed in PBS, and all media were replaced with 10 mL of exosome-free DMEM and incubated for 24 h. The medium was removed and centrifuged at 500× *g* for 10 min to remove cell debris. The supernatant was then filtered through a 0.22 µm filter and ultracentrifuged at 120,000× *g* for 90 min at 4 °C in an SW41 ultracentrifuge rotor (Beckman Coulter, IN, USA) with swing buckets. The supernatant was removed, and the exosome pellet was resuspended in 1 mL filtered PBS and stored at −70 °C until further use.

### 2.4. Exosome Quantification

Exosomes were lysed with RIPA buffer (50 mM Tris pH 6.8, 150 mM NaCl, 1 mM EDTA, 1% NP40) by adding 3:1 volume of RIPA buffer to the exosome sample and mixing thoroughly. The samples were incubated for 30 min on ice and then centrifuged at 13,000× *g* for 5 min at 4 °C. Supernatants were transferred to 1.5 mL Eppendorf tubes and stored at −70 °C. The protein concentration of samples was determined by a Bradford assay. BSA standards at 0.1, 0.25, 0.5, 1, 2, 5 and 10 mg/mL were used as controls. Then, 1 μL of BSA standards were added to designated wells of a 96-well plate in duplicate, and 1 μL of protein samples were loaded in triplicate. Following this, 200 μL of Bradford reagent (Bio-Rad Laboratories, Watford, UK) was added to each well and mixed by pipetting. The plate was incubated at room temperature for 2 min. Absorbance at 595 nm was determined using the Victor3 plate reader (Perkin Elmer, Beaconsfield, UK) and the relative protein concentration of the samples was determined by comparing them to the BSA standards.

### 2.5. Transmission Electron Microscopy

Freshly isolated exosomes in 10 μL PBS were added in a 1:1 ratio with 2% paraformaldehyde and immediately processed for transmission electron microscopy (TEM). Briefly, 5 μL of sample was placed on formvar-coated grids and incubated for 20 min at room temperature. Grids were washed in 100 μL of PBS plus 50 μL of 1% glutaraldehyde for 5 min and then incubated with 100 μL of ddH_2_O for 2 min. Wash steps were repeated eight times in total. After washing, 50 μL of 1% uranyl-oxalate solution (pH 7) was added to the grid for 5 min, then 50 μL of methyl cellulose-UA was added for 10 min on ice. The excess fluid was removed by blotting, and the grids were air dried for 5 to 10 min. Samples were viewed on a JEM-1400 Plus TEM (Jeol, Welwyn, UK) operating at 80 kV. Representative images were collected on an OneView camera (Gatan, Pleasanton, CA, USA). These experiments were carried out at King’s Buildings at The University of Edinburgh.

### 2.6. Nanoparticle Tracking Analysis

Exosomes were analysed by nanoparticle tracking analysis (NanoSight LM10, Malvern Panalytical, Malvern, UK) to determine the size range and distribution. Briefly, 1 mL of diluted exosome sample (1:50–1:100) was loaded on to the NanoSight machine, and particle concentration was determined and diluted in the range of 4 × 10^8^–12 × 10^8^ particles/mL. Parameters were set at a detection rate of 15,000 particles per minute for capture settings, and the smallest vesicle size was set at 30 nm, with analysis performed by NanoSight software version 2.3 (Malvern Panalytical, Malvern, UK). The rate at which exosomes were produced per cell per hour was calculated by dividing the total number of exosomes by the total number of cells after exosome harvesting and then dividing by the number of hours over which the sample was collected.

### 2.7. Exosome Treatment

For exosome treatment, cells were seeded depending on cell type and experimental conditions. Generally, exosomes were added at a concentration of 50 µg/mL. To determine this concentration, 10 μL of isolated exosomes were lysed, and their protein concentration was quantified as in Section 2.4. From that concentration, we calculated the volume of isolated exosomes required to make up a solution at 50 µg/mL in exosome-free media. All exosome solutions were made up fresh prior to treatment. Controls were generated with PBS vehicle instead of exosomes.

### 2.8. Cell Viability Assay

Cells were seeded in 96-well plates at 500 cells per well. Exosomes were added at the indicated concentrations 24 h after seeding. Cell viability was determined 72 h post-treatment using the CellTiter-Glo^®^ Luminescent Cell Viability Assay (Promega, Hampshire, UK) according to the manufacturer’s instructions. Luminescence was measured by a Victor3 multilabel plate reader (Perkin Elmer, Beaconsfield, UK). Data were averaged and normalised against the average signal of the PBS control samples.

### 2.9. Colony Fromation Assay

MDA-MB-231 and MDA-MB-231 RR cell lines were trypsinised and seeded as single cells at 50 cells per well in a 6-well plate. REM134 and REM134 RR were trypsinised and seeded as single cells at 1000 cells per well in a 6-well plate. Immediately after seeding, either PBS (vehicle control), 50 µg/mL exosomes derived from non-RR cells or exosomes derived from RR cells were added to the appropriate well. All plates were incubated as previously described until colonies formed in the vehicle control (approximately 10 days). To stain the colonies, each well was washed with 5 mL PBS and then incubated with 5 mL of 100% methanol for 5 min at room temperature. The methanol was removed, and plates were air dried. Colonies were then stained with a Giemsa stain (20% Giemsa stain (Sigma-Aldrich, Gillingham, UK) plus 80% ddH_2_O) for 20 min at room temperature. After staining, the plates were then washed twice with water and air dried. All colonies were counted and normalised to the control.

In experiments to determine the effect of exosomes derived from RR cells on the colony-forming ability after treatment with radiation, cells were seeded at 20,000 cells in 1 mL of medium in a 12-well plate and incubated for 24 h with either PBS, 50 µg/mL exosomes derived from non-RR cells or 50 µg/mL exosomes derived from RR cells. Cells were then seeded as single cells as described above. In addition, MDA-MB-231 CSCs and MDA-MB-231 RR CSCs were seeded at 750 cells in 3 mL media, and REM134 CSCs and REM134 RR CSCs were seeded at 1000 cells in 3 mL media. Single cells were immediately irradiated at either 0, 2.5 or 5 Gy. Colonies were allowed to form and were processed as described above.

### 2.10. Chemosensitivity Assays

MDA-MB-231 and MDA-MB-231 RR cells were seeded at 500 cells/50 µL per well in a 96-well plate. REM134 and REM134 RR cells were seeded at 1000 cells/50 µL per well in a 96-well plate. CSCs were seeded at 1000 cells/50 µL per well. Cells were incubated for 24 h before treating with 25 µL exosomes (50 µg/mL). Cells were then treated 12 h later with a dose titration of doxorubicin at the indicated concentrations in 25 µL. Cell viability was determined 72 h post-treatment with doxorubicin as described above.

### 2.11. Tumoursphere-Forming Assay

Cells were seeded at 20,000 cells/mL in 1 mL of exosome-free FBS DMEM media in 12-well plates and treated with either PBS, 50 μg/mL of exosomes derived from non-RR cells or 50 μg/mL of exosomes derived from RR cells and incubated for 24 h. Following incubation, MDA-MB-231 and MDA-MB-231 RR cells were seeded as single cells at 3000 cells per well, and REM134 and REM134 RR were seeded at 6000 cells per well, in 3 mL N2 media in 6-well low-attachment plates (Corning, Flintshire, UK). All samples were triplicated. N2 media was supplemented every 48 h with human EFG and human FGF at 10 ng/mL (Peptrotech, London, UK). Sphere formation was monitored for 7 days. Tumourspheres over 50 μm in diameter were counted in five random fields of vision using an Axiovert 40 CFL microscope (Zeiss, Hallbergmoos, Germany) with images taken at 5× and 10× magnification and size measurements recorded by Axiovision software version 4.7.2 (Zeiss, Hallbergmoos, Germany).

### 2.12. Migration Assay

Cells were seeded at 20,000 cells in 1 mL media per well in a 12-well plate and treated with either PBS or corresponding exosomes derived from either non-RR or RR cells at the indicated concentration and incubated for 24 h. Cells were then seeded into Ibidi^®^ (Munich, Germany) chamber slides according to the manufacturer’s instructions. Briefly, cells were trypsinised and seeded at varying concentrations: MDA-MB-231 at 4.5 × 10^5^/well; MDA-MB-232 RR at 4.75 × 10^5^/well; REM134 at 3.45 × 10^5^/well; and REM134 RR at 3.75 × 10^5^/well and incubated until confluent. Once confluent, each insert was removed to leave a gap. Then, 1 mL of media was added to each well and the width of the gap was measured at six points using the Axiovert 40 CFL microscope with an AxioCAM HRm camera (Zeiss, Hallbergmoos, Germany) and pictures were taken at 5× magnification at set time points until the gap was closed. The migration distance was recorded at stated time points with measurements by Axiovision software version 4.7.2. Percentage migration was calculated as (A−B)/B), with A being the size of the gap at 0 h, and B being the gap at the designated time point.

### 2.13. Statistical Analysis

Data were analysed for normality using the Anderson−Darling normality test and the appropriate parametric/non-parametric test was chosen to determine statistical significance. All statistical analyses were performed using Minitab 19 software, with statistical significance being defined as *p* ≤ 0.05.

## 3. Results

### 3.1. Isolation of Exosomes from Canine and Human Breast Cancer Cell Lines and Their Derived RR Counterparts

Radioresistant cell lines MDA-MB-231 RR and REM134 RR were derived by exposing parental cells to increasing doses of radiation every week up until there was limited cell death at 8 Gy [27]. RR cells are morphologically distinct from non-RR parental cells: RR cells have extended cytoplasmic extensions and a spindle-like morphology (Figure 1A(ii,iv)) compared to non-RR cells (Figure 1A(i,ii)). Exosomes were isolated from all cell lines by ultracentrifugation and visualised using TEM. All exosomes exhibited the characteristic “cup shape” morphology [21] (Figure 1B(i–iv)) and expected size distribution as analysed by nanoparticle tracking analysis (NTA) (Figure 1C(i–iv)). NTA was also used to calculate the rate of exosome production per cell per hour and showed that RR cells produced more exosomes than non-RR cells. REM 134 RR cells and MDA-MB-231 RR produced approximately sixfold and threefold more exosomes than their non-RR counterparts, respectively (Figure 1D).

### 3.2. Exosomes Isolated from RR Cells Increased the Survival of Recipient Cells Compared to Exosomes Isolated from Non-RR Cells

To determine the effect of exosomes on cell viability, cells were seeded in 96-well plates, incubated for 24 h and then treated with exosome dilutions of 10, 20, 30, 50 and 75 µg/mL. Cell viability was determined 72 h after treatment. Our data show that exosomes derived from RR cell lines resulted in a significant increase in cell viability, which appeared to be dose dependant, resulting in an increase in cell viability from 100% to 150% (Figure 2A). To compliment the cell viability assay, we also performed colony formation assays. Single cells were immediately treated with either 50 or 100 µg/mL of the corresponding exosomes and incubated until colonies were visible. Exosomes derived from MDA-MB-231 RR and REM134 RR cell lines resulted in a significant increase in the number of colonies compared to both PBS control and exosomes derived from non-RR exosomes (Figure 2B). Based on these results, we selected 50 µg/mL of exosomes to be used in further experiments.

### 3.3. Exosomes Isolated from RR Cells Enhanced the Migration Potential of Recipent Cells

To investigate the effect of exosomes derived from RR cells on the migration potential of REM134 and MDA-MB-231 cells and their RR derivatives, we utilised a 2D scratch assay. Here, cells were incubated with 50 µg/mL of exosomes for 24 h to allow for exosome uptake prior to seeding into a chamber cell with an ibidi insert. Removal of the insert created a defined wound in the cell monolayer. Closure of the wound was measured at the indicated time points until the wound was fully closed (Figure 3). The vehicle control showed that RR cells migrate inherently faster than non-RR cells: non-RR REM134 cells closed the wound 56 h after injury (Figure 3(Ai)) compared to RR REM134 cells, which closed the wound 24 h after injury (Figure 3(Bi)). Similar results, albeit less striking, were obtained for the MDA-MB-231 cell line, whereby non-RR cells closed the wound at 28 h (Figure 3(Ci)) compared to RR cells, which closed the wound at 24 h after injury (Figure 3(Cii)). Exosomes derived from both non-RR and RR cells enhanced the migration potential of recipient cells; however, this effect was more prominent in cells treated with RR exosomes. In non-RR REM134 cells treated with exosomes isolated from non-RR cells, the wound closed at 52 h compared to 48 h for those treated with exosomes derived from RR cells (Figure 3(Ci)). These results were significantly different compared to the control and between treatment groups, such as at 24 h (*p* = 0.0000) for the effect of exosomes derived from RR cells when compared to the control and exosomes derived from non-RR cells. In RR REM134 cells treated with exosomes isolated from non-RR cells, the wound closed at 12 h compared to 8 h for those treated with exosomes derived from RR cells (Figure 3(Cii)). These results were significantly different compared to the control and between treatment groups, such as at 8 h (*p* = 0.0000) for the effect of exosomes derived from RR cells when compared to the control and exosomes derived from non-RR cells. The human cell line showed similar results, in non-RR MDA-MB-231 cells treated with exosomes isolated from non-RR cells, the wound closed at 24 h compared to 12 h for those treated with exosomes derived from RR cells (Figure 3(Ciii)). These results were significantly different compared to the control and between treatment groups, for example at 8 h (*p* = 0.0000) for the effect of exosomes derived from RR cells when compared to the control and exosomes derived from non-RR cells. In RR MDA-MB-231 cells treated with exosomes isolated from non-RR cells, the wound closed at 12 h compared to 8 h for those treated with exosomes derived from RR cells (Figure 3(Ciii)). These results were significantly different compared to the control and between treatment groups such as at the time point of 8 h (*p* = 0.0000) for the effect of exosomes derived from RR cells when compared to the control and exosomes derived from non-RR cells.

### 3.4. Recipient Cells of Exosomes Isolated from Estalished RR Cells Were More Resistant to Chemotherapy and Irradiation Compared to Those Treated with Exosomes from Non-RR Cells

Adjuvant chemotherapy and radiotherapy are commonly used modalities to treat breast cancer in both humans and dogs [22]. Doxorubicin is a common chemotherapeutic used in the treatment of mammary carcinomas [23,24]. To determine the effect of exosomes on the sensitivity of recipient cells to doxorubicin, cells were treated with 50 µg/mL of exosomes isolated from either RR or non-RR cells and incubated for 24 h prior to treatment with the indicated dose titration of doxorubicin. Cell viability was determined 72 h post-treatment (Figure 4A). Exosomes isolated from REM134 RR cells resulted in a significant increase in cell viability of both types of recipient cells, REM134 RR (Figure 4(Ai)) and REM134 non-RR (Figure 4(Aii)) compared to exosomes isolated from non-RR cells and PBS controls, such as at 0.001 µM (*p* < 0.00001) in both the REM134 and the REM134 RR cell line. The exosomes derived from the non-RR MDA-MB-231 cell line did not result in a significant increase in percentage cell viability when compared to the PBS control when added to the MDA-MB-231 cell line (Figure 4(Aiii)), except in the MDA-MB-231 RR cell line (Figure 4(Aiv)) at the concentration of 0.001 µM (*p* < 0.01).

To assay the effect of exosomes isolated from RR cells on the resistance of recipient cells to radiotherapy, we utilised a colony formation assay to assess cell survival and clonogenic growth. Here, non-RR or RR cells were incubated with 50 µg/mL of exosomes isolated from either non-RR or RR cells prior to seeding as single cells at a low density and immediately irradiating at the indicated doses. The number of colonies were counted after 10 days. Exosomes isolated from RR cells significantly increased the colony-forming ability of recipient cells after irradiation at 2.5 and 5 Gy compared to exosomes isolated from non-RR cells or the PBS vehicle control (Figure 4B). This effect was more striking in the non-RR cells treated with exosomes isolated from RR cells in both canine (Figure 4(Bi)) and human (Figure 4(Biii)) cell lines, compared to RR cells treated with exosomes isolated from RR cell lines (Figure 4B(ii,iv)). Our results show that exosomes derived from the RR breast cancer cell lines can alter the phenotype of recipient cells and enhance their resistance to doxorubicin and irradiation.

### 3.5. Exosomes Isolated from RR Cells Can Alter the Phenotype of CSCs

CSCs are inherently more resistant to conventional cancer therapies than surrounding bulk (non-CSC) cancer cells. To determine the effect of exosomes isolated from RR cells on recipient CSCs, we enriched for CSCs using an established tumoursphere assay from all cell lines [28]. CSCs were pre-incubated with exosomes isolated from either RR, non-RR cells or PBS control for 24 h prior to treatment with the indicated dose titration of doxorubicin. Cell viability was assayed 72 h later. Our results show that exosomes isolated from RR cells significantly increased the percentage of cell viability for all recipient CSCs when compared to exosomes isolated from non-RR cells or the PBS vehicle control (Figure 5A). These results were consistent regardless of RR status and both in REM134 cell lines (Figure 5A(i,ii)) and in MDA-MB-231 cell lines (Figure 5A(iii,iv)). We also noted that PBS-treated RR CSCs were inherently more resistant to doxorubicin at all indicated doses than non-RR CSCs, and this was consistent in both cell lines (Figure 5A).

To investigate the effect of exosomes isolated from RR cells on recipient CSCs after radiotherapy, we assayed their colony-forming ability after irradiation. CSCs were pretreated with exosomes for 24 h prior to seeding as single cells at a low density and then immediately irradiated at 0, 2.5 and 5 Gy. The number of colonies were counted after approximately 10 days. Exosomes isolated from the REM134 RR and MDA-MB-231 RR cell lines significantly increased the number of colonies formed and, therefore, the radioresistance of all recipient CSCs compared to treatment with exosomes derived from non-RR cells or the PBS control (Figure 5B). To a much lesser extent, recipient cells treated with exosomes isolated from non-RR cell lines produced relatively more colonies after irradiation treatment compared to the PBS control. This was statistically significant in both non-RR REM134 CSCs (*p* < 0.031 at 2.5 Gy and *p* < 0.003 at 5 Gy) and RR REM134 CSCs (*p* < 0.00001) (Figure 5B(i,ii)) and for non-RR MDA-MB-231 CSCs at 2.5 Gy (*p* < 0.05) (Figure 5(Biii)). Significantly, our results show that exosomes derived from RR cells can change the radioresistance potential of recipient CSCs.

### 3.6. Exosomes Derived from RR Cells Increased the Size of the CSC Pool

To observe the effect of exosomes isolated from RR cells on the tumoursphere-forming ability of recipient cells, REM134, REM134 RR, MDA-MB-231 and MDA-MB-231 RR cells were incubated with 50 µg/mL of exosomes isolated from the indicated cell lines for 24 h, cells were then seeded into low-attachment plates with N2 media to allow the formation of 3D tumourspheres. REM134 and REM134 RR tumourspheres were counted after 5 days. MDA-MB-231 and MDA-MB-231 RR tumourspheres were counted after 17 days. Our results showed that exosomes isolated from both non-RR cell lines (MDA-MB-231 and REM134) and RR cell lines (MDA-MB-231 RR and REM134 RR) significantly increased tumoursphere-forming capacity, both in the number of tumourspheres formed and in the relative size of individual tumourspheres (Figure 6A(i,iv)). Recipient cells of exosomes isolated from non-RR cells produced approximately twice as many tumourspheres compared to the PBS control. This was consistent in all cell lines (Figure 6B(i,iv)). REM134 and REM134 RR recipient cells treated with exosomes isolated from RR cells produced a 3-fold and 4.5-fold increase in tumoursphere formation compared to PBS control, respectively (Figure 6B(i,ii)). Both MDA-MB-231 and MDA-MB-231 RR recipient cells treated with exosomes isolated from RR cells produced approximately 2.5-fold increase in tumoursphere formation compared to PBS control (Figure 6B(iii,iv)). Recipient cells treated with exosomes isolated from RR cells produced significantly larger tumourspheres compared to those receiving exosomes isolated from non-RR cells or the PBS control (Figure 6C(i,iv)). Interestingly, recipient cells treated with exosomes isolated from non-RR cells produced significantly larger tumourspheres compared to the PBS control (Figure 6C(i,iv)). Together, our results indicate that exosomes derived from RR cell types can significantly increase the tumoursphere-forming ability of recipient cells and enhance the overall survival of CSCs, indicating that exosomes derived from RR cell lines may increase the size and hardiness of the CSC pool, and this may drive treatment failure in a clinical setting.

## 4. Discussion

Radiotherapy treatment is critical in the management of human breast cancers, with up to 94% of invasive breast cancer patients receiving radiotherapy treatment plans after surgery in conjugation with chemotherapy [29]. Despite progress made in the precision delivery of radiation and personalised radiotherapy schedules, the development of radioresistance in clinical settings is a significant clinical challenge, which ultimately leads to relapse and metastasis [27]. The tumour microenvironment plays an important role, driving tumour progression and therapeutic response. Exosomes are small extracellular vesicles, containing a large array of active biomolecules that are secreted by different cells into the extracellular matrix of the tumour microenvironment. They are then internalised by recipient cells and then release their content to mediate gene expression and protein activity [12]. Cellular stresses, including radiation and hypoxia, affect exosome secretion, composition, abundance and potential binding to recipient cells [28,29,30,31]. Previous studies have shown that radiation can enhance the release of exosomes and change their molecular composition and that exosomes are capable of transferring radiation-induced effects to non-irradiated cancer cells, therefore, potentially mediating radiation bystander effects [32,33]. Most of these reports have mainly focused on pre- and postradiation changes in exosomal proteins and miRNAs rather than on the mechanisms involved in these changes or their effect on biological functions [30,34,35]. In these studies, exosomes are usually harvested between 1 and 96 h after irradiation treatment [35]. In general, there is a lack of radioresistant model systems to facilitate elucidating the mechanisms underlying the development of radioresistance. In our lab, we previously developed and extensively characterised novel in vitro radioresistant cell lines from human breast cancer (MCF-7, ZR-751 and MDA-MB-231) and canine mammary carcinoma (REM-134) cell lines [26,36]. We found that the radioresistance phenotype was maintained long term, even in the absence of radiation exposure, and concluded that the acquisition of radioresistance was not transient [26]. In this study, we utilised these radioresistant model systems to show that exosomes derived from established RR breast cancer cell lines are capable of changing the phenotype of non-RR recipient cells and inducing radioresistance within 24 h of uptake. Our data suggest that radioresistance is transmittable via exosomes and that, once acquired and established, radioresistance could potentially spread throughout a tumour and beyond. This may be reflective of the observation that any factor affecting the phenotype of a donor cell likely affects the molecular composition of the exosome released by that cell. Our results are consistent with previous studies that investigated the functional role of exosomes in the response of exosomes to radiation exposure. These studies showed that exosomes secreted from head and neck cancer cells within 24 h of irradiation increased the proliferation, survival and migration potential of both non-irradiated and irradiated recipient cells [31,37]. Similarly, exosomes isolated from irradiated glioblastoma cells enhanced the migration phenotype of recipient cells, and molecular profiling revealed an abundance of molecules important for cell migration [38]. However, in these studies, as well as our study, conditioned media collected from irradiated cells prior to exosome isolation were not used as a positive control to confirm that exosomes can mediate this effect within the context of a more complex secretome including other extracellular vesicles.

To date, no studies have mapped changes in exosome composition through the process of acquiring radioresistance. In future studies, we aim to utilise our panel of established RR cell lines to compare the cargo of exosomes derived from RR cells and non-RR cells. Current knowledge in radiation-induced changes in exosome cargo is limited and refers mainly to proteomic changes. There are several studies showing that exosomes derived from irradiated cells can increase the levels of proteins involved in transcription and translation, chaperones, ubiquitin-related proteins and proteosome components and downregulate the proteins associated with response to stress, immunity, cell adhesion and immunity [31,35]. Future research should also focus on the minutiae of exosome uptake and processing to determine what drives the selective uptake of exosomes derived from radioresistant cells/cancer stem cell populations, as it would be beneficial to identify the fate of exosomes derived from radioresistant cells once they are internalised by recipient cells. Do all recipient cells take up exosomes equivalently? Or are subsets of cells primed to take up exosomes secreted by irradiated cells? Can we block this interaction using either small-molecule compound inhibitors or neutralising antibodies? Do all recipient cells respond the same once donor exosomes have been taken up? These are interesting questions that warrant further investigation.

The use of exosomes as a minimally invasive platform for evaluating the circulating biomarkers of a multitude of physiological and pathological processes (including cancer, pregnancy disorders, cardiovascular diseases and immune responses) is gaining traction. Exosomes exist in almost all body fluids and are very stable as they are encapsulated by lipid bilayers, this enhances the clinical applicability of exosomes. Exosomes and their cargo are also representative of parental cells and contain more biological information than cell-free DNA or conventional serum-based biomarkers. Within the context of solid cancers, although solid biopsy is still the gold standard for pathological diagnosis and basis for treatment, the use of serum-based exosomes as biomarkers of cancer has been demonstrated in gliomas [39,40,41], liver cancers [42,43], endometrial cancer [44] and gastrointestinal cancers [45,46]. Exosomes in urine have also been investigated for their possible use in the diagnosis and prognostication of prostate cancer [47,48]. As the production of exosomes and their composition is altered by radiation treatment, exosomes could potentially be used as non-invasive diagnostic markers for radiosensitivity and to monitor the emergence of radioresistance.

Breast cancers are highly heterogeneous and contain a small subset of CSCs. CSCs are inherently more resistant to radiation treatment that non-CSCs and more likely to survive treatment and re-initiate tumour growth [27]. Here, we show that exosomes isolated from RR breast cancer cells have similar effects on both CSCs and non-CSCs, notably conferring resistance to radiation. Interestingly, exosomes isolated from both RR and non-RR cells increased the sphere-forming ability of recipient cells, but this was enhanced by the former significantly more, indicating that exosomes isolated from RR breast cancer cells may increase the size of the CSC pool. We also showed that exosomes isolated from RR cells increased the migratory ability of recipient cells, indicating that that these exosomes activate an EMT, which is associated with cellular plasticity and the acquisition of CSC characteristics [49]. Although, we have shown that exosomes isolated from RR breast cancer cells confer a radioresistance phenotype on recipient cells and that recipient cells have enhanced sphere-forming ability, we have not unequivocally shown that the increased radioresistance is due to an increased proportion of inherently resistant CSCs. Further studies will focus on confirming whether recipient cells of exosomes isolated from RR cells activate an EMT and whether this process is the predominant underlying molecular mechanism driving emerging radiation resistance in naïve cells.

In this study, we compared human and canine breast cancer cells as canine mammary cancer is considered as an excellent translational model of human breast cancer. Naturally occurring mammary tumours are the most frequently diagnosed cancer in bitches, and these tumours represent 50% of all canine tumours, of which 50% are malignant [50]. The main treatment option for dogs is surgery alone due to a lack of receptor status evaluation or molecular subtype classification. Previously, in our lab, we compared the RR REM-134 cell line with a panel of RR human cell lines to investigate the mechanisms of acquired radioresistance and identified a number of similarities including the expression of epithelial and mesenchymal genes and WNT, PI3K and MAPK pathway activation [26]. Here, we demonstrate that exosomes isolated from human and canine RR cell lines have similar functional effects on recipient cells and that the process of potentiating exosome-mediated radioresistance is comparable in humans and dogs. We believe that a “One Health” approach is crucial to unpick tumourigenesis and to develop future treatment strategies that will benefit both species.

## 5. Conclusions

Our study provides compelling evidence that exosomes can serve as an effective communication tool in the development of radioresistance and can confer pro-survival signals and promote the radioresistant phenotype to non-radioresistant cells. This study indicates a functional role for exosomes within our models in the dissemination of aggressive cancer characteristics. Further studies are required to map the cargo of exosomes derived from RR cells and to identify and validate potential therapeutic targets to halt the perpetuation of acquired radioresistance throughout a tumour.

## Figures and Tables

**Figure 1 jpm-11-01310-f001:**
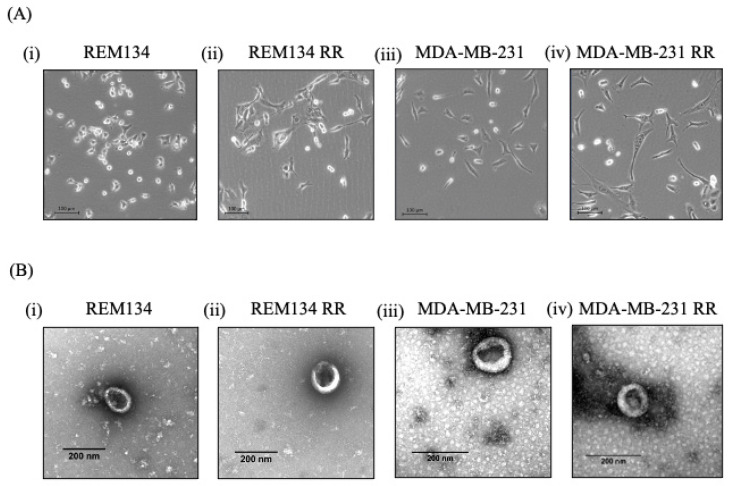
Isolation of exosomes from canine and human breast cancer cell lines and their derived isogenic RR counterparts. (**A**) Cell morphology of (**i**) REM134, (**ii**) REM134 RR, (**iii**) MDA-MB-231 and (**iv**) MDA-MB-231 RR cells. Scale bar represents 100 μm. (**B**) Visualisation, using TEM, of exosomes isolated from (**i**) REM134, (**ii**) REM134 RR, (**iii**) MDA-MB-231 and (**iv**) MDA-MB-231 RR cells. Scale bar represents 200 nm. Characterisation of exosomes using NTA to measure (**C**) particle distribution from (**i**) REM134 cells, (**ii**) REM134 RR, (**iii**) MDA-MB-231 and (**iv**) MDA-MB-231 RR and (**D**) rate of exosome production per cell per hour. Data are representative of three independent experiments.

**Figure 2 jpm-11-01310-f002:**
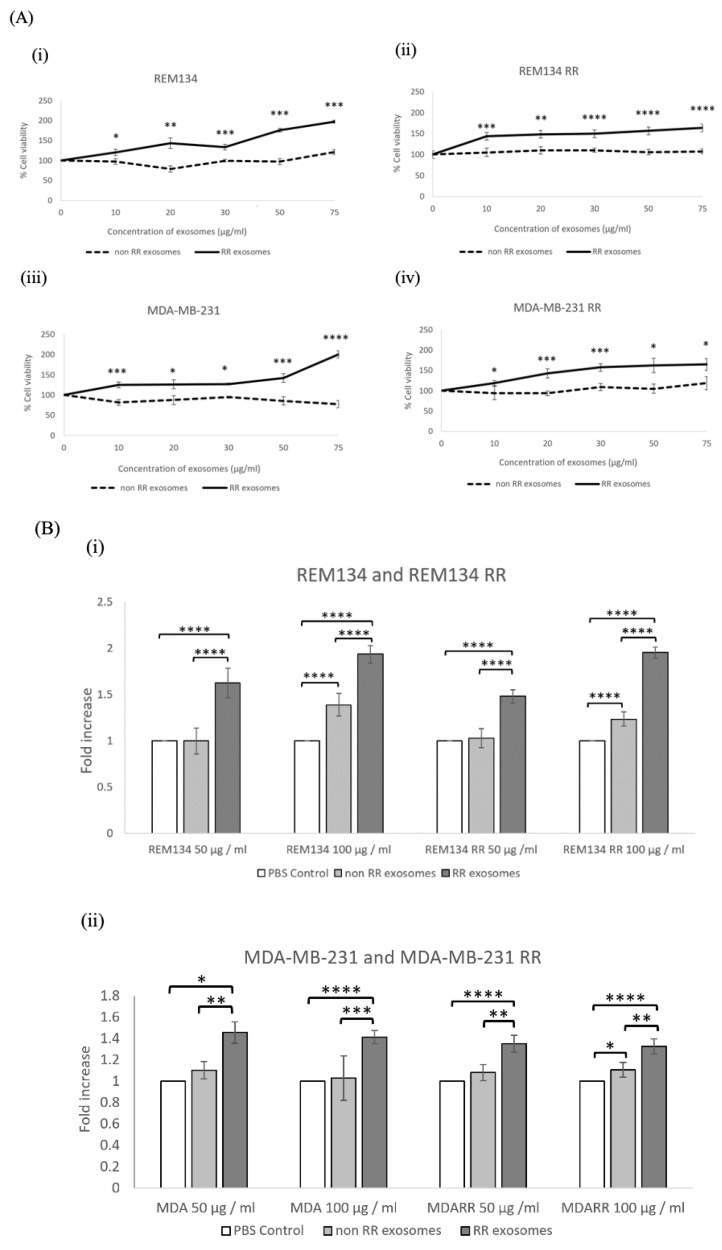
Exosomes isolated from RR cell lines increased the survival of recipient cells. Analysis of (**A**) cell viability and (**B**) colony-forming ability were assayed after (**i**) REM134, (**ii**) REM134 RR, (**iii**) MDA-MB-231 and (**iv**) MDA-MB-231 RR cells were treated with the indicated dose of exosomes isolated from either corresponding non-RR or RR cells. All results are relative to the appropriate PBS control. Three repeats were performed and analysed by a two-sample *t* test. Error bars indicate ±SD. * *p* ≤ 0.05, ** *p* ≤ 0.01; *** *p* ≤ 0.001, **** *p* ≤ 0.00001.

**Figure 3 jpm-11-01310-f003:**
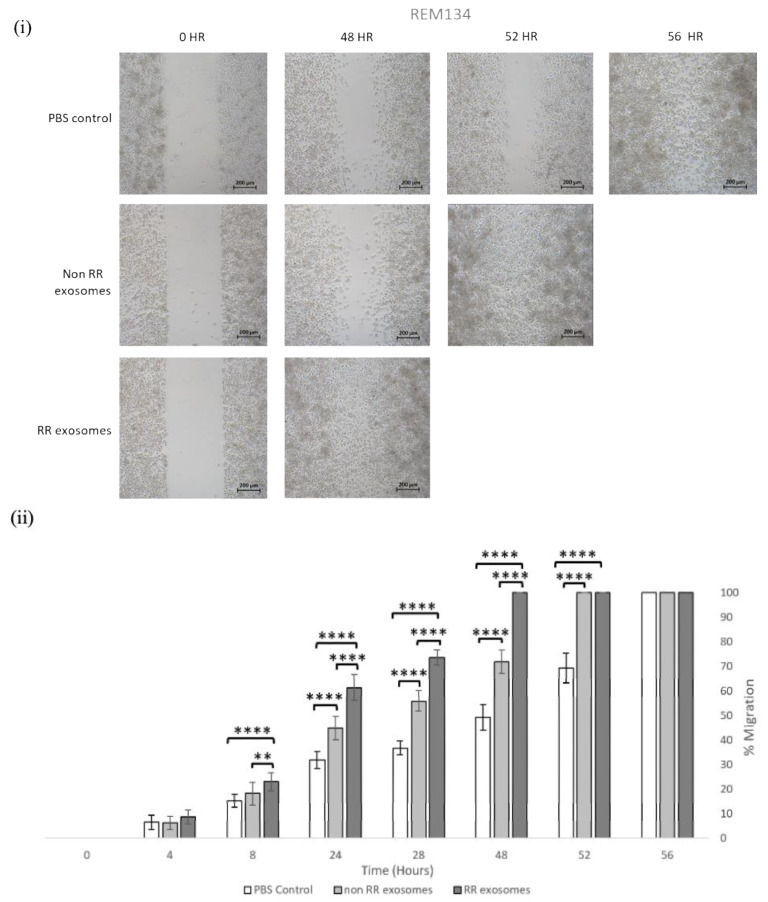
RR cells migrate faster than non-RR cells, and the migration potential in all cell types was enhanced after treatment with exosomes isolated from RR cell lines. Migration potential was assayed by an in vitro wound-healing assay in (**A**) REM134, (**B**) REM134 RR, (**C**) MDA-MB-231 and (**D**) MDA-MB-231 RR cells. The indicated cell line was treated with either PBS or exosomes isolated from either non-RR (50 μg/mL) or RR corresponding cells (50 μg/mL). (**i**) Light microscopy images of cell migration at the indicated time points are shown. (**ii**) Graphical representation of relative migration compared to the PBS control at the indicated time points. Three biological repeats were performed, and a two-sample *t* test was used for the analysis of data. Error bars indicate ±SD. ** *p* ≤ 0.01; *** *p* ≤ 0.001, **** *p* ≤ 0.00001.

**Figure 4 jpm-11-01310-f004:**
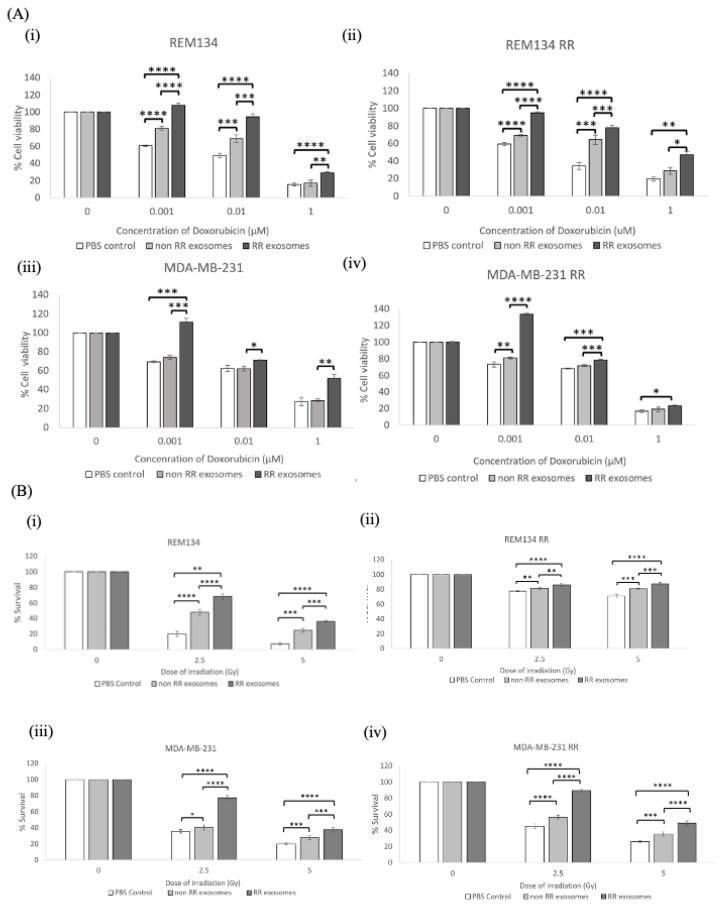
Exosomes isolated from RR cells increased the resistance of recipient cells to doxorubicin and ionising radiation. (**A**) Chemosensitivity to increasing doses of doxorubicin was determined for (**i**) REM134, (**ii**) REM134 RR, (**iii**) MDA-MB-231 and (**iv**) MDA-MB-231 RR cells. Cells were seeded for 24 h with exosomes (50 μg/mL) isolated from either non-RR or RR corresponding cell lines prior to treatment with the indicated dose of doxorubicin. Cell viability was assayed 72 h after doxorubicin treatment. (**B**) Colony-forming ability after treatment with 0, 2.5 or 5 Gy was determined for (**i**) REM134, (**ii**) REM134 RR, (**iii**) MDA-MB-231 and (**iv**) MDA-MB-231 RR cells. All cell lines were pretreated with exosomes (50 μg/mL) isolated from either non-RR or RR corresponding cell lines for 24 h prior to irradiation. Three repeats were performed, and significance was determined by a two-sample *t* test. Error bars indicate ±SD. * *p* ≤ 0.05, ** *p* ≤ 0.01; *** *p* ≤ 0.001, **** *p* ≤ 0.00001.

**Figure 5 jpm-11-01310-f005:**
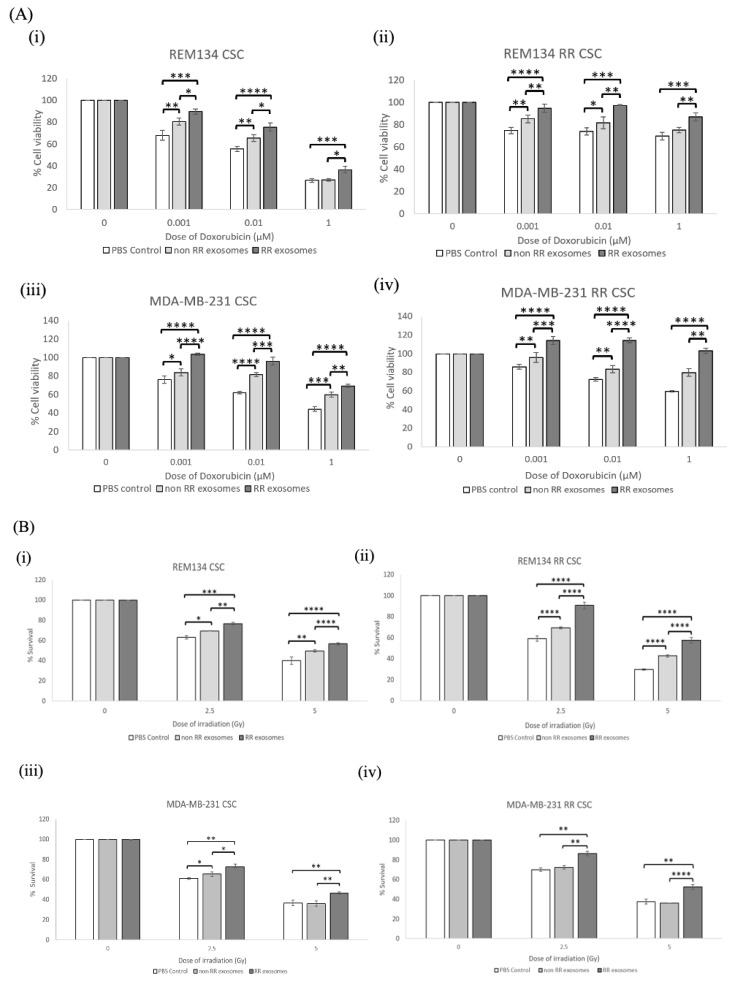
Exosomes isolated from RR cells can alter the resistant phenotype of CSCs. (**A**) Chemosensitivity to increasing doses of doxorubicin was determined for (**i**) REM134, (**ii**) REM134 RR, (**iii**) MDA-MB-231 and (**iv**) MDA-MB-231 RR CSCs. CSCs were pretreated for 24 h with 50 μg/mL exosomes isolated from either non-RR or RR corresponding cell lines prior to treatment with the indicated dose of doxorubicin. Cell viability was assayed 72 h after doxorubicin treatment. (**B**) Colony-forming ability after treatment with 0, 2.5 or 5 Gy was determined for (**i**) REM134, (**ii**) REM134 RR, (**iii**) MDA-MB-231 and (**iv**) MDA-MB-231 RR CSCs. All CSCs were pretreated with 50 μg/mL of exosomes isolated from either non-RR or RR corresponding cell lines for 24 h prior to irradiation. Three repeats were performed, and data were analysed by a two-sample *t* test. Error bars indicate ±SD. * *p* ≤ 0.05, ** *p* ≤ 0.01; *** *p* ≤ 0.001, **** *p* ≤ 0.00001.

**Figure 6 jpm-11-01310-f006:**
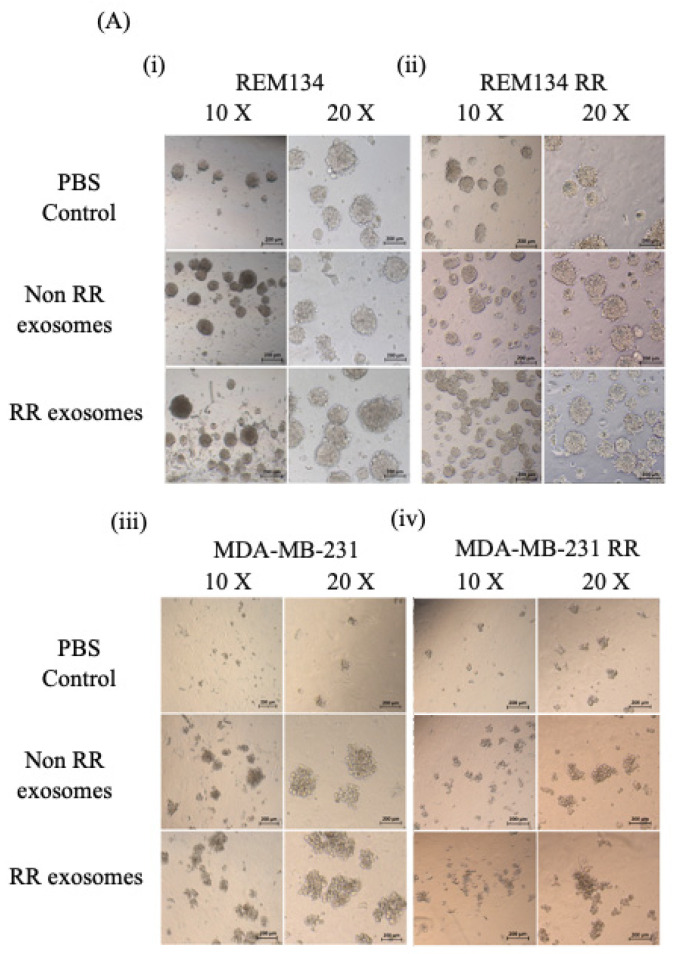
Exosomes isolated from RR cells enhanced sphere-forming ability. Spheres were characterised by (**A**) cell morphology, (**B**) number of spheres and (**C**) size of spheres. (**i**) REM134, (**ii**) REM134 RR, (**iii**) MDA-MB-231 and (**iv**) MDA-MB-231 RR cells were treated with 50 μg/mL exosomes isolated from either non-RR or RR corresponding cell lines for 24 h prior to setting up the sphere assay. REM134 and REM134 RR spheres were grown for 7 days, and MDA-MB-231 and MDA-MB-231 RR spheres were grown for 17 days prior to analysis. Three repeats were performed, data were analysed by a two-sample *t* test and size data was analysed by a Wilcoxon signed rank test. Error bars indicate ±SD. **** *p* ≤ 0.00001.

## Data Availability

Not applicable.

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
