# Peer review of "Exosomes Derived from Radioresistant Breast Cancer Cells Promote Therapeutic Resistance in Naïve Recipient Cells"

_jpm, 2021, doi:10.3390/jpm11121310_

Round 1

Reviewer 1 Report

In the manuscript by Payton and co-workers data are presented that suggest that exosomes derived from radioresistant breast cancer cells have the potential to induce a stronger effect on certain biological activities of cancer cells than exosomes derived from radiosensitive parental cells. This was shown with the canine breast cancer cell line REM134 and with the human breast cancer cell line MDA-MB-231.

Though the data are interesting, their physiological relevance is questionable.

First, the concentrations of isolated exosomes the authors have chosen to use are not justified by any analysis of the naturally occurring exosome concentration. Second, conditioned medium (CM) from RR cells and non-RR cells should have been compared first, before studying the contribution of the exosomes on a potential CM effect. Third, except for the measurement of the effect of exosomes on cell viability in 2D cultures, only one (the second highest) concentration was used for studying the biological effects of the exosomes.

Fourth, why did the authors use the mesenchymal MDA-MB-231 cell line, which belongs to a rare breast cancer subgroup? As mentioned in the discussion section, the authors have generated RR sublines from ERalpha-positive MCF-7 and ZR-75-1 cell lines, which would have been much more appropriate for this study. In addition, MDA-MB-231 cells are genetically very unstable. Radio-resistant sublines may have additional features unrelated to RR, which may play a role in the effects of their exosomes. For that reason, at least another RR-MDA-MB-231 subline should have been tested. Furthermore, MDA-MB-231 cells are highly motile, have a high content of CSCs and have a high metastatic activity. Showing that exosomes increase the aggressiveness of an already aggressive cell line is not really important. Again, it would have been much better to show this with MCF-7 and/or ZR-75-1 cells. Also, it is odd that the authors mention in the introductory section that CSCs are typically CD44-positive and CD24-negative and then use the MDA-MB-231 cell line where the majority of the cells, if not all do show these features and CSCs in this cell line are defined by ALDH activity.

Further comments

Fig. 2A/B: How do the authors explain that RR cells benefit from RR-cell-derived exosome to the same extent as non-RR cells? Wouldn’t RR cells not be exposed to their own exosomes they produce constantly? What is the concentration of exosomes produced by RR cells after 72 hours compared to the concentration of exosomes added?

Fig. 3C, lines 315/316: The effect of RR-cell derived exosomes on migration is minimal: the time for wound healing reduced from 52 only to 48h at the second highest concentration of exosomes. Do lower concentrations, which may be closer to physiological ones, have also an effect?

Fig. 5: When 3000-6000 cells per well are seeded, the chance is high that the cells aggregrate rather than form colonies from single cells. The authors do not demonstrate that they have actually analyzed CSCs. The results just show that RR cell-derived exosomes have an effect on doxorubicin-treated cancer cells in 3D cultures, without having any indication whether their stemness activity has changed.

Fig. 6: Again, how can the author be sure the colonies were formed from single cells? If floating cells are not prevented from moving, e.g. by embedding them in agarose, random aggregation of cells occur which then grow up to spheroids. Hence, the higher number of spheroids in the presence of RR cell-derived exosomes might just be explained by higher aggregation and/or motility activities.

minor issues

lines 31/32: The statement “Breast cancer is …the second leading cause of cancer-related deaths in women [1,2].” is not correct. It is the leading cause. This is also confirmed by the data in reference 2 (figure 4C),

Fig. 2B, 3-6: Legends are too small and barely readable even after enlargement. Pictures with higher resolution are desirable.

typos

line 57: tumourigensis

line 73: nucleuic

Author Response

Comments and Suggestions for Authors
In the manuscript by Payton and co-workers data are presented that suggest that exosomes derived from radioresistant breast cancer cells have the potential to induce a stronger effect on certain biological activities of cancer cells than exosomes derived from radiosensitive parental cells. This was shown with the canine breast cancer cell line REM134 and with the human breast cancer cell line MDA-MB-231.
Though the data are interesting, their physiological relevance is questionable.
Thank you for reviewing our paper and for your constructive comments. We have tried to address the comments below in red.
First, the concentrations of isolated exosomes the authors have chosen to use are not justified by any analysis of the naturally occurring exosome concentration.
Thank you for your comment. We do have data showing the number of exosomes produced by each cell type and we have added it into the paper as figure 1 D.

To the Materials and Methods section, we added line 187 - 190:
“Rate of exosomes produced per cell per hour was calculated by dividing the total number of exosomes by the total number of cells after exosome harvesting and then dividing by the number of hours over which the sample was collected.”

Added to Results line 267 to 271:
“NTA was also used to calculate the rate of exosome production per cell per hour and showed that RR cells produced more exosomes than non-RR cells. REM 134 RR cells and MDA-MB-231 RR produced approximately 6-times and 3-times more exosomes than non-RR counterparts, respectively (Figure 1 D).”

From that data you can see that RR cells produce more exosomes than non-RR cells. This is independent of proliferation as we have previously shown that REM 134 RR cells proliferate slower than non- RR cells (Gray et al (2020) Comparative analysis of the development of acquired radioresistance in canine and human mammary cancer cell lines. Frontiers in Vet. Sci. doi: 10.3389/fvets.2020.00439). In this paper we were mainly comparing the effects of exosomes isolated from RR cells to non-RR cells and therefore concluded that it was more comparable to use the same dose of exosomes for all cell types rather than different native
conditions.

We selected the concentration of 50 µg/ml for all experiments beyond figure 2 based on the cell viability results (as presented in figure 2). Here, we used a dose titration of exosomes from 10 – 75 µg/ml, although all doses tested increased cell viability, a dose of 50 µg/ml had the most profound effect on all cell lines.

We have added the following line to the text (line 295 - 296):
“Based on these results we selected 50 µg/ml of exosomes to be used in further
experiments.”

Second, conditioned medium (CM) from RR cells and non-RR cells should have been compared first, before studying the contribution of the exosomes on a potential CM effect.

Ongoing work in our lab is characterising the secretome of RR cells. In this study we focused on the contribution of exosomes to mediate the RR phenotype and did not deem it necessary to conduct the experiments with conditioned media first.

Third, except for the measurement of the effect of exosomes on cell viability in 2D cultures, only one (the second highest) concentration was used for studying the biological effects of the exosomes.

Please find our justification for using this dose above.

Fourth, why did the authors use the mesenchymal MDA-MB-231 cell line, which belongs to a rare breast cancer subgroup? As mentioned in the discussion section, the authors have generated RR sublines from ERalpha-positive MCF-7 and ZR-75-1 cell lines, which would have been much more appropriate for this study. In addition, MDA-MB-231 cells are genetically very unstable. Radio-resistant sublines may have additional features unrelated to RR, which may play a role in the effects of their exosomes. For that reason, at least another RR-MDA-MB-231 subline should have been tested. Furthermore, MDA-MB-231 cells are highly motile, have a high content of CSCs and have a high metastatic activity. Showing that exosomes increase the aggressiveness of an already aggressive cell line is not really important. Again, it would have been much better to show this with MCF-7 and/or ZR-75-1 cells. Also, it is odd that the authors mention in the introductory section that CSCs are typically CD44-positive and CD24-negative and then use the MDA-MB-231 cell line where
the majority of the cells, if not all do show these features and CSCs in this cell line are defined by ALDH activity.

Thank you for your comment. We chose to use the MDA-MB-231 cell line (ER− /PR−/HER2−) as it exhibits a mesenchymal-like phenotype similar to REM 134 cells. We also classified REM 134 cells as ER− /PR− /HER2+. We therefore concluded that these two cell lines were more comparable than MCF-7 and ZR-751 cell lines (both classified as ER+/PR+/HER2−) especially as MCF-7 and ZR-751 RR derivatives lost ER and PR expression, becoming ER− /PR− /HER2−, whereas no change was identified in receptor expression between the parental and RR MDA-MB-231 or REM 134 cell lines.

We appreciate that MDA-MB-231 cell line is very aggressive but argue that if our results are significant in this already aggressive cell line then the underpinning biological mechanism driving exosome-dependent acquisition of radiation resistance is fundamental and not transient depending on disease progression.

We have removed the line from the introduction referring to breast cancer stem cells as typically CD44 positive and CD24 negative.

Further comments
Fig. 2A/B: How do the authors explain that RR cells benefit from RR-cell-derived exosome to the same extent as non-RR cells? Wouldn’t RR cells not be exposed to their own exosomes they produce constantly? What is the concentration of exosomes produced by RR cells after 72 hours compared to the concentration of exosomes added?

Thank you for your comment. We appreciate this point and posed the same questions when we produced this data. Firstly, our results are consistent with those of Mutschelknaus et. al. (2016, 2017) that showed that exosomes isolated from irradiated head and neck squamous cell carcinoma cells were able to increase the proliferation and survival of both non-irradiated and irradiated cancer cells.

We have included a discussion of these points in the paper (line 521 – 532):
“Our data suggests that radioresistance is transmittable via exosomes and that, once acquired and established, radioresistance could potentially spread throughout a tumour and beyond. This may be reflective of the observation that any factor affecting the phenotype of a donor cell likely affects the molecular composition of the exosome released by that cell. Our results are consistent with previous studies that investigated the functional role of exosomes in the response of exosomes to radiation exposure. These studies showed that exosomes secreted from head and neck cancer cells within 24 hours of irradiation increases the proliferation, survival and migration potential of both non-irradiated and irradiated recipient cells [31,37]. Similarly, exosomes isolated from irradiated glioblastoma cells enhanced the migration phenotype of recipient cells and molecular profiling revealed an abundance of molecules important for cell migration [38].”

Secondly, we hypothesise that RR cells may take up their own exosomes to maintain their own radioresistance phenotype. This would be akin to how cancer cells induce autocrine signalling to become self-sufficient in growth factor signalling by producing and taking up their own growth factors such as EGFR.

What is the concentration of exosomes produced by RR cells after 72 hours compared to the concentration of exosomes added?

This is a difficult question to answer, mainly because the concentration of exosomes at any given time is in constant flux as exosomes are not only being produced by cells but also being taken up by cells. These cells are also not synchronised and different cell populations at different stages of the cell cycle will be producing and taking up exosomes at different rates. This was one of the main reasons that we tried to standardise experiments by using 50 µg/ml concentration of exosomes rather than the native conditions that differed between cell types.

As an estimate we isolated exosomes from 12 ml of media taken from a confluent T75 flask. Exosome pellet was resuspended in 1 ml PBS. We lysed 10 µl to determine protein concentration and from that we calculated the volume of exosomes required to make up 50 μg/ml in exosome-free media prior to treating cells. As a very rough estimate we calculate this to be 40,000,000 exosomes.

Published papers that have used a similar concentration range to us include:

Kim, J. Y., Rhim, W.-K., Yoo, Y.-I., Kim, D.-S., Ko, K.-W., Heo, Y., Park, C. G., & Han, D.K. (2021). Defined MSC exosome with high yield and purity to improve regenerative activity. Journal of Tissue Engineering. https://doi.org/10.1177/20417314211008626

Lukic, A., Wahlund, C., Gómez, C., Brodin, D., Samuelsson, B., Wheelock, C. E.,
Gabrielsson, S., & Rådmark, O. (2019). Exosomes and cells from lung cancer pleural exudates transform LTC4 to LTD4, promoting cell migration and survival via CysLT1. Cancer letters, 444, 1–8. https://doi.org/10.1016/j.canlet.2018.11.033

Lee J-K, Park S-R, Jung B-K, Jeon Y-K, Lee Y-S, Kim M-K, et al. (2013) Exosomes Derived from Mesenchymal Stem Cells Suppress Angiogenesis by Down-Regulating VEGF Expression in Breast Cancer Cells. PLoS ONE 8(12): e84256.
https://doi.org/10.1371/journal.pone.0084256

Hu, Z., Yuan, Y., Zhang, X., Lu, Y., Dong, N., Jiang, X., Xu, J., & Zheng, D. (2021). Human Umbilical Cord Mesenchymal Stem Cell-Derived Exosomes Attenuate Oxygen-Glucose Deprivation/Reperfusion-Induced Microglial Pyroptosis by Promoting FOXO3a-Dependent Mitophagy. Oxidative medicine and cellular longevity, 2021, 6219715.https://doi.org/10.1155/2021/6219715

We have added the following section to the Materials and Methods:
“2.7. Exosome treatment

For exosome treatment cells were seeded depending on cell type and experimental conditions. Generally, exosomes were added at a concentration of 50 µg/ml. To determine this concentration, 10 μl of isolated exosomes were lysed and protein concentration quantified as in section 2.4. From that concentration we calculated the volume of isolated exosomes required to make up a solution at 50 µg/ml in exosome-free media. All exosome solutions were made up fresh prior to treatment. Controls were performed with PBS vehicle instead of exosomes.”

Fig. 3C, lines 315/316: The effect of RR-cell derived exosomes on migration is minimal: the time for wound healing reduced from 52 only to 48h at the second highest concentration of exosomes. Do lower concentrations, which may be closer to physiological ones, have also an effect?

Unfortunately, we don’t have this data. This experiment was conducted three times at 50 μg/ml only. The question of what is a physiologically relevant dose of exosomes is a complex one and one must take into account that within a solid tumour what proportion of cancer cells are radioresistant and producing RR exosomes, and that the proximity to non-RR cells will determine the dose received, whereby those close by will get a much higher dose than those
further away. Replicating those diffusion gradients is a very interesting biological question but beyond the scope of this study.

Fig. 5: When 3000-6000 cells per well are seeded, the chance is high that the cells aggregrate rather than form colonies from single cells. The authors do not demonstrate that they have actually analyzed CSCs. The results just show that RR cell-derived exosomes have an effect on doxorubicin-treated cancer cells in 3D cultures, without having any indication whether their stemness activity has changed.

Thank you for your comment. We have previously characterised CSCs isolated from the REM 134 cell line. This work has been published here:
Pang, L.Y., Cervantes-Arias, A., Else, R.W., and Argyle, D.J. (2011). Canine
Mammary Cancer Stem Cells are Radio- and Chemo- Resistant and Exhibit an
Epithelial-Mesenchymal Transition Phenotype. Cancers. 3 (2), 1744-62. 

MDA-MB-231 CSCs have been characterised here:
Hiraga, T., Ito, S., & Nakamura, H. (2011). Side population in MDA-MB-231
human breast cancer cells exhibits cancer stem cell-like properties without higher bone-metastatic potential. Oncology Reports, 25, 289-296.
https://doi.org/10.3892/or_00001073

Here, we argue that enhanced chemoresistance and radioresistance in CSCs compared to nonCSC in the PBS controls is evidence of an increase in CSC characteristics.

Fig. 6: Again, how can the author be sure the colonies were formed from single cells? If floating cells are not prevented from moving, e.g. by embedding them in agarose, random aggregation of cells occur which then grow up to spheroids. Hence, the higher number of spheroids in the presence of RR cell-derived exosomes might just be explained by higher aggregation and/or motility activities.

We appreciate the point but there is an increase in the size of the spheroids compared to the PBS control this combined with the data presented in figure 5 where we show that there is an increase in resistance to chemotherapy and radiotherapy solidifies that the sphere assay is enriching for CSCs and is not an experimental artifact caused by aggregation.

minor issues
lines 31/32: The statement “Breast cancer is …the second leading cause of cancer-related deaths in women [1,2].” is not correct. It is the leading cause. This is also confirmed by the data in reference 2 (figure 4C),

Thank you. We have updated this sentence to:
“Breast cancer is the most common female malignancy and the leading cause of cancerrelated deaths in women”

Fig. 2B, 3-6: Legends are too small and barely readable even after enlargement. Pictures with higher resolution are desirable.

We increased the size of figure 2B. All legends are size 9 font according to the journal style guidelines.

typos line 57: tumourigenesis
Corrected
line 73: nucleuic
Corrected

Reviewer 2 Report

The manuscript is well written and can be

accepted  for publication.

Author Response

The manuscript is well written and can be accepted for publication.
Thank you for your kind and positive review. 

Reviewer 3 Report

Payton and colleagues present a paper aimed at analyzing whether radioresistant cell-derived exosomes can promote radioresistance in radiosensitive cells. Overall, this is an exciting paper; it offers solid, novel data and is well-written. I do have several concerns to be addressed before publication.

The morphological differences between radioresistant and radiosensitive cells beg the question: did the number of exosomes vary between radioresistant and radiosensitive cells? Please discuss.

Please clarify what is to be understood as 'concentration of exosomes' from Figure 2 on. The methods section describes a protein quantitation method under Exosome quantification (lines 151-164) and an exosome count under Nanoparticle tracking analysis (lines 180-188). So, were the experiments conducted using as many exosomes to reach a given protein concentration or using a given amount of exosomes? This difference does not seem trivial as the study focuses on the effects of exosomes.

Along similar lines, please discuss the rationale behind the selection of 50 µg/mL for the experiments presented in Figure 2B and beyond. 

The discussion section can be substantially improved. Regarding its structure,  lines 475-502 would be better suited to the introduction as it only provides data. The results from this paper are re-stated twice, in lines 503-508 and 574-578, as is the group's previous experience with radioresistant cells (lines 498-501 and 579-582). 

Understandably, it is challenging to compare such novel findings with those of other groups, but I think that, besides the role of exosomes as biomarkers, the authors can further discuss these findings in the context of resistance acquisition, an aspect only marginally addressed in lines 507-508.

Minor points

The roman numerals in figures add unnecessary complexity since the figures in each panel (e.g., Figure 1A, B, etc.) are clearly labeled and rarely cited in the manuscript individually. I suggest removing them for a cleaner look. Additionally, the size of the labels in Figure 3 should be increased slightly.

The images in Figure 6a are labeled 5x and 10x magnification. Considering the apparent size of the cells in the images, I think that 5x and 10x are the employed objectives and that the actual magnification is in the 50-100x range. Please clarify.

Author Response

Comments and Suggestions for Authors
Payton and colleagues present a paper aimed at analyzing whether radioresistant cell-derived exosomes can promote radioresistance in radiosensitive cells. Overall, this is an exciting paper; it offers solid, novel data and is well-written. I do have several concerns to be addressed before publication.
Thank you for your thoughtful review and for raising such interesting points. All of your comments have been addressed below in red. 

The morphological differences between radioresistant and radiosensitive cells beg the question: did the number of exosomes vary between radioresistant and radiosensitive cells? Please discuss.
Thank you for your comment. We do have this data and have added it into the paper as figure1 D.

To the Materials and Methods section, we added line 187 - 190:

“Rate of exosomes produced per cell per hour was calculated by dividing the total number of exosomes by the total number of cells after exosome harvesting and then dividing by the number of hours over which the sample was collected.”

Added to line 267 to 271:
“NTA was also used to calculate the rate of exosome production per cell per hour and showed that RR cells produced more exosomes than non-RR cells. REM 134 RR cells and MDA-MB-231 RR produced approximately 6-times and 3-times more exosomes than non-RR counterparts, respectively (Figure 1 D).”

Please note that the increased rate of exosome production in the RR cells is independent of increased proliferation as we have previously shown that REM 134 RR cells proliferate slower than non- RR cells (Gray et al (2020) Comparative analysis of the development of acquired radioresistance in canine and human mammary cancer cell lines. Frontiers in Vet. Sci. doi: 10.3389/fvets.2020.00439).

Please clarify what is to be understood as 'concentration of exosomes' from Figure 2 on. The methods section describes a protein quantitation method under Exosome quantification (lines 151-164) and an exosome count under Nanoparticle tracking analysis (lines 180-188). So, were the experiments conducted using as many exosomes to reach a given protein concentration or using a given amount of exosomes? This difference does not seem trivial as
the study focuses on the effects of exosomes.

Thank you for your comment. Experiments were conducted using as many exosomes to reach a given concentration. We lysed a proportion of the harvested exosomes and assayed for protein concentration and from that concentration we calculated the volume of our harvested exosome preparation required to make up 50 μg/ml in exosome-free media prior to treating
cells.

We had added the section has been added to the Materials and Methods:
“2.7. Exosome treatment
For exosome treatment, cells were seeded depending on cell type and
experimental conditions. Generally, exosomes were added at a concentration of 50  µg/ml. To determine this concentration, 10 μl of isolated exosomes were lysed and protein concentration quantified as in section 2.4. From that concentration we calculated the volume of isolated exosomes required to make up a solution at 50 µg/ml in exosome-free media. All exosome solutions were made up fresh prior to treatment. Controls were performed with PBS vehicle instead of exosomes.”

Along similar lines, please discuss the rationale behind the selection of 50 µg/mL for the experiments presented in Figure 2B and beyond.
In the cell viability assay presented in figure 2 we used a dose titration of exosomes from 10– 75 µg/ml, although all doses tested increased cell viability, a dose of 50 µg/ml had the most profound effect on all cell lines and we decided to use that dose going forward in future experiments.

We have added the following line to the text (line 295 - 296):
“Based on these results we selected 50 µg/ml of exosomes to be used in further
experiments.”

The discussion section can be substantially improved. Regarding its structure, lines 475-502 would be better suited to the introduction as it only provides data. The results from this paper are re-stated twice, in lines 503-508 and 574-578, as is the group's previous experience with radioresistant cells (lines 498-501 and 579-582).
Thank you for the suggestion. I appreciate that the discussion is a bit long and I have tried to streamline it by removing some of the superfluous information including the following two paragraphs:

Abramowicz et. al. (2019) isolated exosomes from head and neck cancer cell lines 24 hours after irradiation at 2, 4 or 8 Gy and found that 369 proteins increased in abundance at all radiation doses and were predominantly associated with cellular response to radiation including metabolism of radical oxygen species, DNA repair, chromatin repackaging and protein folding. The authors concluded that “…the protein content of exosomes released by irradiated cells indicates their actual role in mediating the response to ionizing radiation” [33].

For example, An et. al. (2017) isolated serum exosomes from patients with locally advanced pancreatic cancer over a course of chemotherapy and identified specific changes in the levels of 8 proteins associated with metastasis and resistance to therapy [50]. Kulkarni et. al. (2016) performed proteomic analysis of urine and serum exosomes of patients undergoing radiotherapy and found that urine derived exosomes can be used to detect radiation damage in the liver and gastrointestinal systems [51]. Collectively, these reports confirm that exosomes may serve as noninvasive diagnostic biomarkers for radiosensitivity and radioresistance, and to monitor the emergence of radioresistance. 

However, I respectfully disagree that lines 475 – 502 should be moved to the introduction. I think that this section is an important discussion of the studies that have looked at the emergence of radioresistance from a different perspective to us and gives context to our results and why we have conducted our study this way.

I also appreciate the point that the summary of results is re-stated twice but the second time is in relation to using dogs as a translational model of humans and to highlight similarities we identified between humans and dogs in our previous work. I therefore believe that both summaries are justified and should be included. 

Understandably, it is challenging to compare such novel findings with those of other groups, but I think that, besides the role of exosomes as biomarkers, the authors can further discuss these findings in the context of resistance acquisition, an aspect only marginally addressed in lines 507-508.
Thank you for the comment. We added the following (line 521 – 532) and three additional references:
“Our data suggests that radioresistance is transmittable via exosomes and that, once acquired and established, radioresistance could potentially spread throughout a tumour and beyond. This may be reflective of the observation that any factor affecting the phenotype of a donor cell likely affects the molecular composition of the exosome released by that cell. Our results are consistent with previous studies that investigated the functional role of exosomes in the response of exosomes to radiation exposure. These studies showed that exosomes secreted from head and neck cancer cells within 24 hours of irradiation increases the proliferation, survival and migration potential of both non-irradiated and irradiated recipient cells [31,37]. Similarly, exosomes isolated from irradiated glioblastoma cells enhanced the migration phenotype of recipient cells and molecular profiling revealed an abundance of molecules important for cell migration [38].”  

Minor points

The roman numerals in figures add unnecessary complexity since the figures in each panel (e.g., Figure 1A, B, etc.) are clearly labeled and rarely cited in the manuscript individually. I suggest removing them for a cleaner look.
Thank you for your comment but we respectfully disagree with the above change as this is a style preference and our preference is to label all subfigures.

Additionally, the size of the labels in Figure 3 should be increased slightly.
We have increased the size of figure 3.

The images in Figure 6a are labeled 5x and 10x magnification. Considering the apparent size of the cells in the images, I think that 5x and 10x are the employed objectives and that theactual magnification is in the 50-100x range. Please clarify.

Thank you for your comment. There was an oversight here: 5x and 10x is the size of the objectives but there is a 2x magnification between the lens of the microscope and the camera, giving an effective magnification of 10x for the 5x lens and 20x for the 10x lens. We have updated the labels of figure 6 A to reflect this. 

Round 2

Reviewer 1 Report

The authors addressed most of my concerns satisfactorily, although I still think that experiments with conditioned media would have been an important positive control. By having only studied exosomes, the authors do show that exosomes have the potential to induce RR, but not if this activity is relevant in the complex effect of conditioned medium. In other words, does conditioned medium containing exosomes from RR cells induce RR? At least, this point should be brought up in the Discussion section.

I am also still not convinced that the authors analyzed CSC activity by the spheroid assay they have performed. Unless the authors show by other means that CSC activity was affected by RR exosomes, I would strongly advice to remove “cancer stem cell populations” from the title, so now reading “Exosomes derived from radioresistant breast cancer cells promote therapeutic resistance in naïve recipient cells”
